# Primary Lymphoma of the Gynecologic Tract: A Comprehensive Pathologic Analysis of 15 Cases

**DOI:** 10.3390/diagnostics15081016

**Published:** 2025-04-16

**Authors:** Haneen Al-Maghrabi, Jaudah Al-Maghrabi

**Affiliations:** 1Department of Pathology and Laboratory Medicine, King Faisal Specialist Hospital and Research Center, Jeddah 23433, Saudi Arabia; 2Department of Pathology, Faculty of Medicine, King Abdulaziz University, Jeddah 23433, Saudi Arabia; jalmaghrabi@hotmail.com

**Keywords:** Burkitt lymphoma, diffuse large B-cell lymphoma, immunohistochemistry, primary gynecologic lymphoma

## Abstract

**Background:** Diagnoses of primary gynecologic (GYN) lymphomas are rare and require a high clinical index of suspicion, with only case reports and case series being presented. The aim of this study is to assess and record the pathological distinction of primary GYN lymphomas at two tertiary hospitals. **Materials and Methods:** We retrieved all cases diagnosed from 2000 to 2024, reviewed histopathology slides, and performed additional extra immunohistochemistry staining for required cases. In addition, follow-up information available up to the point of writing this study was gathered. **Results:** The records of two hospitals indicated 15 instances of primary GYN lymphoma in patients aged between 18 and 79. Only one patient was human immunodeficiency virus (HIV)-positive. Symptoms in most cases were non-specific, and most of the patients presented with the primary complaint of unexplained bleeding. A total of 13 cases were classified as diffuse large B-cell lymphoma (DLBCL), 1 as acute B-lymphoblastic lymphoma (B-LBL), and 1 as Burkitt lymphoma (BL). **Conclusions:** Primary lymphoma of the GYN tract is a rare disease, with DLBCL being the most prevalent type in our area. Our review, based on experiences from two large tertiary centers in the western region of Saudi Arabia, aims to contribute to this effort. The reporting of additional cases of primary GYN lymphoma is needed, which may aid in a better understanding of its pathogenesis as well as enhancing diagnosis and treatment outcomes.

## 1. Introduction

It is extremely uncommon to find non-Hodgkin’s lymphoma (NHL) originating primarily in the gynecologic (GYN) region, with the primary involvement of the GYN tract only observed in 0.2% to 1.1% of cases [1,2]. It has been estimated that approximately 165 instances of primary lymphoma in the pelvic or genital tract are diagnosed each year in the United States [3,4]. According to the existing literature, the estimated range of female genital tract lymphomas occurring as part of the disseminated or secondary involvement of the GYN tract is 7% to 30%, with an estimated total of around 2500 cases [2,3,5]. Late-stage dissemination of the disease to the uterus and ovaries is commonly observed in autopsies of women with advanced lymphoma, indicating the late involvement of these organs [6,7]. B- and T-cell lymphomas can be diagnosed in the GYN tract, with additional classification into respective subgroups. The B-cell phenotype is the most common, accounting for 80% of all lymphomas in this area. Traditional treatments yield better response rates and prognoses for B-cell lymphomas, compared to T-cell lymphomas [2,3]. NHL affecting the female reproductive system involves the occurrence of lymphomas in regions such as the uterine body and cavity, ovaries, cervix, vagina, and vulva. Either primary or secondary involvement can be used to classify lymphomas that affect the adjacent pelvic organs. The differentiation between primary and secondary involvement holds clinical significance due to crucial variations in associated prognoses and treatment modalities. The early detection of primary GYN lymphoma allows for prompt treatment with chemotherapy and/or radiation therapy, preventing the need for surgery which can impede the treatment’s progress. Clinical and radiologic imaging can play a crucial role in distinguishing lymphomas from the more prevalent malignant conditions affecting the pelvic region [6]. It is commonly understood that lymphomas typically originate from regions where there is a significant accumulation of lymphoid tissue, such as the lymph nodes, spleen, and bone marrow [3]. The physiologic circulation ability of lymphocytes allows for the potential occurrence of primary tumors in different, “unexpected” parts of the body. The identification of primary GYN lymphoma can be aided by radiologic imaging and pathologic tissue examination, along with bone marrow biopsy. Unfortunately, it becomes clinically and pathologically challenging to differentiate between primary and secondary lymphomas once the disease has spread. There is controversy surrounding whether normal lymphoid tissue exists in both the uterus and ovaries [8]. Monterroso Et Al. detected a low count of lymphocytes inside the ovaries, encompassing blood vessels in the hilus and inside or around the corpora lutea in 1993, confirming Woodruff Et Al.’s discoveries from 1963 [5]. The results of this study lend credence to the idea that, although uncommon, the development of malignant lymphomas in GYN organs is possible.

## 2. Materials and Methods

A retrospective analysis of hematoxylin and eosin (H&E)-stained pathology slides obtained from the archives of two main tertiary centers in the western region of Saudi Arabia (King Abdulaziz University Hospital and King Faisal Specialist Hospital and Research Centre, Jeddah, Saudi Arabia) was performed. All pathology slides were processed and reviewed in accordance with the updated 2022 classification of tumors in hematopoietic and lymphoid tissues by the World Health Organization (WHO). Through a morphological review incorporating supplementary immunohistochemistry markers, the reclassification of previous cases into currently accepted diagnostic categories was conducted by two lymph node and lymphoma-specialized pathologists. All cases diagnosed as primary GYN lymphoma within the period from 2000 to 2024 were considered as part of the inclusion criteria. Cases with secondary GYN involvement and cases without pathology slides or paraffin blocks were not considered in our study, as part of the exclusion criteria. The definition for primary extranodal NHL proposed by Krol et al. [9] was adopted, which encompasses all patients with NHL originating from any GYN organ—even in cases with disseminated disease, if the GYN organ component is the primary clinical feature. The collected research data included different parameters, such as the patient’s age during presentation, the affected location, the pathology subtype, the treatment provided, and the outcome at follow-up. Information regarding the clinical and pathological aspects of the patients was gathered from their records. The minimal required set of immunohistochemistry ancillary tests done on paraffin embedded tissue using a BOND III automated immunostainer (Leica Biosystems, Nussloch, Germany), included CD45 (DAKO, Jena, Germany), CD20, PAX-5 (Leica Biosystems), CD5, CD3, BCL-2, BCL-6, CD10, MUM1, Cyclin-D1, CD23, CD15, CD30, and the Ki-67 proliferation index. These markers were accurately analyzed while ensuring that internal controls were maintained. To rule out other potential diagnoses, additional markers such as pan cytokeratin, OCT-4, SALL-4, desmin, HMB-45, PAX-8, and S-100 were used, depending on the location of the tumor. In cases of B-lymphoblastic lymphoma/leukemia, additional markers such as CD79a, CD19, CD22, TdT, MPO, CD1a, CD117, and those from relevant flow cytometry studies were also taken into consideration. Selected cases of high-grade morphology were sent for additional extra tests, including fluorescence in situ hybridization (FISH) for MYC reciprocal translocation, as well as partner genes t (8; 14) (q24; q32) associated with c-MYC-IGH. Moreover, one case was sent for in situ hybridization (ISH) to Epstein–Barr virus (EBV)-encoded RNA (EBER) (Ventana, Export, PA, USA), to confirm the diagnosis of BL (as discussed below). Approval for the study was granted by the Research Committee of the biomedical ethics unit at King Abdulaziz University Hospital (Reference No. 34-22) and the research ethics committee at King Faisal Specialist Hospital and Research Center, Jeddah, Saudi Arabia (IRB-2024-CR-31). Table 1 presents the data that were gathered.

## 3. Result

Our study identified 15 cases of primary GYN lymphoma, with a summary of their clinicopathological attributes provided in Table 1. The diagnoses for all these cases were established through histopathologic examinations submitted to our pathology department. Of the total number of cases, eight were diagnosed using biopsies and the others were diagnosed after resection. Frozen sections were performed for three of these surgical resection cases. Two were diagnosed during frozen interpretation as atypical lymphoid proliferation highly suspicious for lymphoproliferative disorders, while the third case was diagnosed as high-grade malignant neoplasm. Most of these patients exhibited non-specific symptoms, such as bleeding in nine patients (60%), while the others showed signs of a lesion causing mass effect (40%). The patients’ ages ranged between 18 and 79 years, with a median age of 44 years and a mean age of 49 years. One patient had a history of bone marrow (BM) transplant; later, she developed pre-B-acute lymphoblastic leukemia (B-ALL), with no lymphoma at that time. She was treated with complete remission. Two years after her treatment, she presented with a huge right ovary and tubal mass; her BM was normal during these investigations. Four patients presented with primary endometrial/uterine involvement (26%), five individuals with cervical lesions (33%), and six with ovarian mass (40%), two of which presented adjacent fallopian tube involvement. Furthermore, 1 patient was diagnosed with acute B-lymphoblastic lymphoma (B-LBL) (6%; Figure 1A,B); 1 with Burkitt lymphoma (BL) (6%), which had previously been published [10] (Figure 1C,D); and 13 with DLBCL (86.6%; Figure 1E,F). None of these patients were detected to have BM involvement at the time of GYN lymphoma diagnosis. COG0232 protocol-based consolidation was administered to the patient for treatment of B-LBL and, after follow-up for 6 years, no signs of disease recurrence were observed. The case diagnosed with BL completed four cycles of R-CODOXM/R-IVAC, presenting no disease recurrence for 6 years post treatment. Cases diagnosed with DLBCL were treated with R-CHOP, with no disease recurrence observed at follow-up for 3 to 12 years in eight patients (53.3%), while one patient showed only partial response to treatment (6.6%), likely due to her disease progression involving multiple mesenteric, para-aortic lymph nodes, and pericardial involvement. Unfortunately, her follow-up lasted for only 3 years. Sadly, one patient passed away four months after her initial diagnosis; she was an HIV-positive case with two adnexal masses. Despite the fact that HIV-positive patients are generally linked to a significantly higher risk of lymphoma development, most of them were aggressive B-cell neoplasms. The exact pathogenic association between primary GYN involvement and decreased immunity is not yet well understood. This could not be further investigated or explained in our study, based on there only being one relevant patient in the total study population (see Table 1). The results of our investigations suggest that several factors can affect the overall survival rate, including tumor histology, the stage of the disease, bone marrow involvement, and overall patient performance.

## 4. Discussion

Lymphoma affecting the GYN tract is a rare diagnosis that poses difficulties for both clinicians and pathologists in terms of diagnosis. As abnormal vaginal bleeding is the most common presentation in affected patients, it is usual practice for a gynecologist to conduct the initial evaluation and treatment. Asymptomatic presentation or experiencing vague abdominal symptoms, such as bloating, discomfort, and pressure, are commonly reported by patients. In another series, only 17% of patients with ovarian lymphomas reported experiencing constitutional symptoms typically observed in those with NHL, such as fever, night sweats, and weight loss [6]. Radiologic studies have been evaluated as a diagnostic tool, due to the vague presenting symptoms. Although ultrasound or computed tomography (CT) scans may not provide specific details regarding solid gynecologic masses, magnetic resonance imaging (MRI) possesses exceptional contrast resolution capabilities that allow for the visualization of underlying organ zonal patterns. Lymphoma appears dark on T1-weighted scans and typically has mild to moderate signal intensity on T2 scans. Moreover, the mass is usually hypovascular if IV contrast is given. Ovarian lymphoma is typically characterized by the appearance of a substantial ovarian mass, measuring approximately 10 cm on average. Bilateral ovarian homogeneous masses of significant size indicate the likelihood of ovarian lymphoma as a possible diagnosis. In over 50% of cases, the disease can progress to an advanced stage and spread to the peritoneal cavity, making it challenging to differentiate it from more prevalent epithelial ovarian tumors [11]. If a pathologic diagnosis of lymphoma through tissue is confirmed, it is essential to commence a comprehensive staging assessment. It is recommended to undergo a complete body CT and/or positron emission tomography (PET) scan, then to proceed with a bone marrow biopsy. PET scanning has become a frequently utilized diagnostic tool for determining the disease stage, detecting any signs of recurrence, and monitoring the effectiveness of treatments [12]. Additionally, CA-125 levels may prove to be advantageous in certain cases; in particular, a decrease in 5-year survival rate has been linked with a high concentration of CA-125 (above 35 U/ml) in patients diagnosed with GYN lymphomas [13]. An advanced clinical stage, the presence of extranodal involvement, bulky mass, the presence of B-symptoms, effusions in pleural and peritoneal cavities, positive bone marrow, elevated serum LDH along with beta2 microglobulin levels, and poor overall patient performance after treatment have all been found to be associated with a higher serum level of CA-125.

DLBCL is the predominant type of NHL that has been observed to occur localized in the uterus. It is also the most common disseminated lymphoma that affects the uterus, accounting for 50% of such instances. Most lymphomas affecting the cervix are confined to its stroma, while tumors impacting the corpus tend to have a higher stage. A false-negative cytologic smear may occur as cervical lymphomas generally have their origin in cervical stroma, causing the superficial squamous epithelium to remain intact [14]. The typical method for diagnosing primary cervical lymphoma includes a deep cervical biopsy, which is then subjected to pathologic evaluation. According to Harris et al. [15], 67% of these tumors exhibited a sub-epithelial mass without any visible signs of ulceration or abnormality in the squamous epithelium lining. If the initial biopsy fails to provide a diagnosis or indicates the presence of atypical epithelial cells along with lymphoid infiltrates, it is advised (by the authors) to perform a deep incisional or excisional cervical biopsy to obtain a definitive diagnosis. The growth pattern of primary cervical lymphomas is characterized by rapid progression. According to one study, despite having normal pelvic examinations within the previous year, six patients had been found to have a cervical lesion measuring 6 cm or greater [16]. In a review conducted by Muntz et al. [17] it was observed that, among a group of patients diagnosed with stage IE cervical lymphomas, half of the women had tumors that exceeded 4 cm at the time of initial diagnosis. Many medical centers tend to prefer combined therapy even though patients with non-bulky lesions and localized stage IE disease can usually be treated with surgery, chemotherapy, or radiotherapy alone. For women with advanced disease, the recommended treatment option is a combination of chemotherapy and customized radiotherapy [16]. Vaginal NHL that is primary or localized, as well as secondary involvement or a high stage, is typically characterized by the presence of both a vaginal mass and bleeding. A vast majority of cases belong to the DLBCL classification. The survival rate for those patients with localized vaginal lymphoma is typically excellent. Older individuals tend to be afflicted with disseminated disease incorporating the vagina, which is unsurprisingly associated with a comparatively shorter survival rate. A literature review revealed 19 patients that had previously been reported to have developed primary vaginal NHL. Vaginal bleeding was the most prevalent presentation, while a vaginal mass was the most frequent clinical finding. The most common histologic type was DLBCL. The follow-up period ranged from 7 months to 20 years, with an 88% 5-year survival rate reported [15,18,19,20]. Another study from France presented a report on six cases of primary vaginal NHL, consisting of five patients with DLBCL and one patient with Burkitt’s lymphoma. All patients underwent chemotherapy, with some additionally receiving radiotherapy. After an average follow-up of 5.7 years, five patients remained alive while one patient passed away due to unrelated reasons [21]. Primary vulvar NHL has been recorded in eight cases in the literature with ages ranging from 25 to 79; among them, six presented with a vulvar mass, while all had a labial mass. Of the six cases assessed, four had tumors of DLBCL lineage while the remaining two were T-cell lymphomas. Among the eight patients, five were treated with chemotherapy and four of them passed away in less than a year [22,23,24,25,26]. Notably, DLBCL of the vulva is more prevalent than mycosis fungoides (MF), despite MF being the typical lymphoid neoplasm that affects the skin, considering that the vulva is classified as a skin site [27]. The most probable reason for predilection in the selection of biopsy sites is that patients with MF typically have affected skin in many other easily accessible areas for the clinician, which increases the likelihood of those areas being biopsied. Primary lymphomas discovered solo in fallopian tubes are extremely rare; however, the fallopian tubes are often affected by primary lymphomas that originate in adjacent ovaries. A deep analysis of the existing literature revealed that primary lymphoma of the fallopian tubes is highly unusual. One scenario identified in the literature suggested a potential occurrence of primary fallopian tube lymphoma [28], in which a 68-year-old woman was diagnosed with hydrosalpinx along with tubo-ovarian adhesion. A gross pathologic examination revealed greatly thickened and swollen fimbriae ends. The authors reported the expansion of the fimbriated end caused by low-grade follicular lymphoma (FL). There was no further follow-up available regarding that case. Due to the systemic nature of FL, we cannot fully accept that this instance originated solely from the fallopian tube without further affirming data. 

From a pathology standpoint, the diagnosis and sub-classifications of GYN lymphomas are determined through their morphology and immunophenotype. These tumors have extremely heterogeneous patterns, architecture, and nuclear features. Understanding their pathologic classification is very important, as it has an effect on the corresponding treatment. Diffuse large B-cell lymphoma and Burkitt’s lymphoma are commonly diagnosed types of GYN high-grade lymphomas, which grow and spread very quickly. Other types of lymphoma—such as follicular lymphoma, marginal zone lymphoma, and lymphoplasmacytic lymphoma—tend to be more slowly progressing types. Chemotherapy is very effective in treating NHL tumors that divide rapidly [29]. At present, the standard first-line therapy for lymphoma is a combination of chemotherapy drugs called R-CHOP. This combination is typically given for at least three cycles, but may be extended to six cycles for patients with bulky GYN disease. The combination of chemotherapy and radiation has an estimated cure rate of about 60–70% [30]. Notably, treatment with Bendamustine–Rituximab extended the free survival time, increased the rate of complete disease response, and caused fewer side effects in patients with indolent NHL than treatment with R-CHOP [31]. 

In summary, GYN and pelvic primary lymphomas can be difficult to diagnose as they tend to present similar signs and symptoms to other gynecological malignancies, such as endometrial, cervical, ovarian carcinomas, carcinosarcoma, or sarcomas. As such, primary lymphoma at these locations is a very rare diagnosis that requires a high clinical index of suspicion. The decision regarding which patients to biopsy can be challenging, as there are many other, more common causes of pelvic masses. Diagnostic radiologic examination and intraoperative pathology of frozen sections can be helpful in making a diagnosis, but they are not always definitive. The lack of clarity regarding the exact diagnosis could result in a patient undergoing unnecessary surgery. Depending on the extent of the disease and the patient’s condition, the patient can be treated non-surgically. Chemotherapy is the preferred treatment for GYN and pelvic lymphomas, and surgical staging along with debulking are not necessary. A biopsy of a target lesion before surgery can help to obtain a more accurate diagnosis.

## Figures and Tables

**Figure 1 diagnostics-15-01016-f001:**
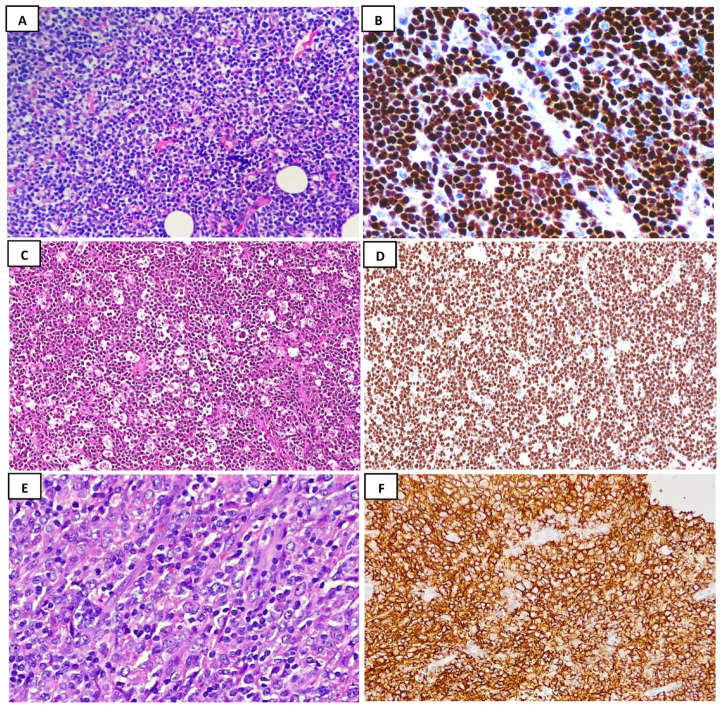
Hematoxylin and eosin (H&E)-stained slides along with their contributory immunohistochemistry. (**A**) B-lymphoblasts with an open chromatin pattern infiltrating fibroadipose tissue in a case of B-lymphoblastic lymphoma (B-LBL) (H&E; ×20); (**B**) B-LBL expresses a strong nuclear TdT staining pattern (40×); (**C**) histopathologic examination reveals diffuse lymphoid infiltrate characterized by numerous cells that are arranged in sheets and exhibit a “starry-sky” appearance, consistent with Burkitt lymphoma (BL) (H&E; ×20); (**D**) the Ki-67 proliferative index is almost 100% in the BL case (×20); (**E**) sheets of large atypical B-lymphoid cells with vesicular chromatin and prominent nucleoli, presenting diffuse infiltrate compatible with diffuse large B-cell lymphoma (DLBCL) (H&E; ×40); (**F**) a DLBCL case showing sheets of diffuse membranous CD20-positive B-cells (×20).

**Table 1 diagnostics-15-01016-t001:** Clinicopathologic Features of 15 Cases of Primary GYN Lymphoma.

	Age	Medical History	Tumor Size (cm)	Site of Involvement	Diagnosis	Immunohistochemistry Stains Results	Follow-Up (Years)	Treatment	Outcome
1	63	Pelvic mass adherent to the vagina. Another mass in the anterior aspect of the cervix.	1st mass 7.9 × 5.5 2nd mass 7.5 × 5	Cervical and Vaginal	DLBCL (Non-GCB)	Positive: CD20, PAX-5, CD79a, MUM1. Negative: CD10, BCL-6, CD5, CyclinD1. Ki-76 is 70%	5	R-CHOP	No recurrence
2	34	HIV patient with two adnexal masses displacing the bowel and extending to both ureters.	12 × 11 and 10 × 9	Adnexal and Pelvic Mass	DLBCL (Non-GCB)	Positive: CD20, PAX-5, MUM1, BCL-6. Negative: CD10, CD5, CD3, CD23, CyclinD1. Ki-76 is 80%	4 months	R-CHOP	Patient died 4 months after diagnosis
3	69	Pelvic left adnexal mass.	7.2 × 6.4	Left Ovary	DLBCL (Non-GCB)	Positive: CD20, PAX-5, CD79a, MUM1, BCL-6. Negative: CD10, CD5, CD3, CD23, CyclinD1. Ki-76 is 60%	3	R-CHOP	No recurrence
4	33	Left pelvic adnexal mass with multiple mesenteric, paraaortic lymph node enlargement, peritoneal involvement, and pericardial mass.	NA	Left Ovary	DLBCL (Non-GCB)	Positive: CD20, PAX-5, CD79a, MUM1. Negative: CD10, BCL-6, CD5, CyclinD1. Ki-76 is 70%	3	R-CHOP	Partial response
5	61	Uterine mass.	6.4 × 6	Uterus	DLBCL (Non-GCB)	Positive: CD20, PAX-5, MUM1, BCL-6. Negative: CD10, CD5, CD3, CD23, CyclinD1. Ki-76 is 60%	NA	No available clinical follow-up to data	NA
6	68	Pelvic mass infiltrating the uterus.	NA	Uterus	DLBCL (Non-GCB)	Positive: CD20, PAX-5, CD79a, MUM1, BCL-6. Negative: CD10, CD5, CD3, CyclinD1. Ki-76 is 70%	3	R-CHOP	No recurrence
7	44	Pelviabdominal mass infiltrating the bladder, sigmoid, and broad ligament.	NA	Cervix and Fallopian Tubes	DLBCL (Non-GCB)	Positive: CD20, PAX-5, MUM1, BCL-6. Negative: CD10, CD5, CD3, CyclinD1. Ki-76 is 60%	NA	NA	NA
8	39	Uterus with omentum involvement.	18 × 16 × 13	Uterus	DLBCL (Non-GCB)	Positive: CD20, PAX-5, CD79a, MUM1. Negative: BCL-6, CD10, CD5, CD3, CD23, CyclinD1. Ki-76 is 70%	12	R-CHOP	No recurrence
9	44	Stage IVB lung and uterine, BM negative.	13.4 × 9.9 × 8.2	Uterus	DLBCL (Non-GCB)	Positive: CD20, PAX-5, CD79a, MUM1. Negative: BCL-6, CD10, CD5, CD3, CD23, CyclinD1. Ki-76 is 70%	8	R-CHOP	No recurrence
10	79	Advanced disease and poor performance status.	7.5 × 6.5 × 6	Cervical Mass/Uterus Involvement	DLBCL, (GCB)	Positive: CD20, PAX-5, CD79a, BCL-6, CD10. Negative: MUM1, CD5, CD3, CD23, CyclinD1. Ki-76 is 70%	NA	Palliative for supportive care	No show/patient lost contact
11	18	BMT, Pre-B-ALL case with complete remission. Two years after treatment presented with huge pelvic mass, BM-negative.	20 × 15	Right Ovary and Right Fallopian Tube	B-ALL/LBL	Positive: CD19, CD79a, CD22, TdT, CD34, CD10 and PAX-5. Negative: Myeloperoxidase and lysozyme.	6	Right ovary and right fallopian tube resection + COG0232 protocol-based consolidation complete	No recurrence
12	47	NA.	Right 12 × 10 and left 15 × 10	Bilateral Ovaries and Tubes	BL	Positive: CD19, CD20, CD79a, PAX-5, CD10, BCL6, MYC, EBV (ISH). Negative: BCL-2, CD3, CD5. Ki67 is ~ 100%	6	Bilateral oophorectomy + completed 4 cycles of R-CODOXM/R-IVAC	No recurrence
13	29	NA.	NA	Cervical Mass	DLBCL, (Non-GCB)	Positive: CD20, PAX-5, CD79a, MUM1. Negative: BCL-6, CD10, CD5, CD3, CD23, CyclinD1. Ki-76 is 70%	3	6 cycles of R-CHOP	No recurrence
14	39	Stage IVB.	16 × 15	Left Ovary	DLBCL, (GCB)	Positive: CD20, PAX-5, CD79a, BCL-6, CD10. Negative: MUM1, CD5, CD3, CD23, CyclinD1. Ki-76 is 70%	3	Surgical resection + 6 cycles of R-CHOP and 3 cycles of prophylactic MTX	No recurrence
15	68	Uterus/cervix involvement, and lung Stage IV.	13 × 9 × 8.9	Cervical Mass	DLBCL, (Non-GCB)	Positive: CD20, PAX-5, MUM1. Negative: BCL-6, CD10, CD5, CD3, CD23, CyclinD1. Ki-76 is 70%	3	6 cycles of R-CHOP	No recurrence

DLBCL: Diffuse large B-cell lymphoma; GCB: germinal-center B-cell-like; B-ALL/LBL: B-acute lymphoblastic leukemia/lymphoma; BL: Burkitt lymphoma; BM: bone marrow; BMT: bone marrow transplant; EBV (ISH): in situ hybridization (ISH) to Epstein–Barr virus (EBV)-encoded RNA (EBER); R-CHOP: cyclophosphamide, doxorubicin, prednisone, rituximab, and vincristine; R-CODOXM/R-IVAC: cyclophosphamide, vincristine, doxorubicin, high-dose methotrexate/ifosfamide, etoposide, high-dose cytarabine; MTX: Methotrexate; NA: not available.

## Data Availability

The original contributions presented in this study are included in the article. Further inquiries can be directed to the corresponding author.

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
