# Peer review of "Primary Lymphoma of the Gynecologic Tract: A Comprehensive Pathologic Analysis of 15 Cases"

_diagnostics, 2025, doi:10.3390/diagnostics15081016_

Round 1
Reviewer 1 Report (New Reviewer)
Comments and Suggestions for Authors
This is a retrospective review of 15 cases of primary lymphoma of gyn. tract. In the materials and methods section the authors should explain whether central independent pathology reading was (retrospectively) conducted for all cases? Which imaging methods were used during diagnostic process? When reporting data on rare histologies the authors should discuss a possible role of prospective international databases.
Author Response
Reviewer Comment: This is a retrospective review of 15 cases of primary lymphoma of gyn. tract. In the materials and methods section the authors should explain whether central independent pathology reading was (retrospectively) conducted for all cases? Which imaging methods were used during diagnostic process? When reporting data on rare histologies the authors should discuss a possible role of prospective international databases.
Replay: All pathology slides were prepared and examined following the revised 2022 classification of tumors in hematopoietic and lymphoid tissues established by the World Health Organization (WHO). The review involved assessing morphological features and using additional immunohistochemistry markers slides by two pathologists specializing in lymph nodes and lymphoma; whom are the main authors of this manuscript.
The imaging studies conducted included both CT scans and MRI scans. While the results for all cases were unclear and not specific to a lymphoma diagnosis, we have detailed the findings from both the CT and MRI in the text.
We conducted a thorough literature review of all previously published cases of primary gynecological lymphoma. Despite the limited number of cases, we compared all study parameters with our findings from the retrospective analysis.

Reviewer 2 Report (New Reviewer)
Comments and Suggestions for Authors
I really appreciate the authors idea about the study but I would like to make minor review comments like as they have reported in their study that they also check the presence of different markers like CD20, PAX-5, CD79a, BCL-6, CD10, MUM1, CD5, CD3, CD23, CyclinD1, and Ki-67 in these tumor samples so I would like to see their IHC results. Moreover, I would suggest the author add scale bars in the IHC images for better clarity and analysis and I would appreciate if they also included the negative control in their study (IHC experiments) to confirm the non-specificity of the Antibody they used!
Author Response
Reviewer Comment: I really appreciate the authors idea about the study but I would like to make minor review comments like as they have reported in their study that they also check the presence of different markers like CD20, PAX-5, CD79a, BCL-6, CD10, MUM1, CD5, CD3, CD23, CyclinD1, and Ki-67 in these tumor samples so I would like to see their IHC results. Moreover, I would suggest the author add scale bars in the IHC images for better clarity and analysis and I would appreciate if they also included the negative control in their study (IHC experiments) to confirm the non-specificity of the Antibody they used!
Replay: Thank you very much for the generous review. For each case, whether a biopsy or a resection, we performed an extensive panel of immunohistochemistry. Each panel consists of a minimum of six to ten immune stains. Due to space constraints in the images, we focused on showcasing the most representative histology and key immunohistochemistry stains, such as a case of BL highlighting the Ki-67 proliferative index. We aimed to feature the positive IHC results rather than the negative ones. Nonetheless, all immunohistochemistry stain results are thoroughly detailed in Table 1.
We added a scale bar in the last two images of the present group.
The cases involved resections and biopsies. The biopsies primarily contained tumor tissue, with no normal tissue or entrapped parenchyma present. The resection specimens were entirely infiltrated by tumor, and unfortunately, some revealed no residual parenchymal tissue, such as in the case of bilateral ovarian BL. This situation creates a challenge in capturing images that accurately represent the area of interest. We made every effort to photograph the edges or periphery, but unfortunately, the resolution was poor due to a high degree of cautery artifact. We did our best to present a detailed and accurate description of the cytomorphological aspects of the case, aiming for clarity for the reader.

This manuscript is a resubmission of an earlier submission. The following is a list of the peer review reports and author responses from that submission.
Round 1
Reviewer 1 Report
Comments and Suggestions for Authors
The Authors report a series of cases of non-breast implant lymphomas that were diagnosed as primary breast lymphomas between 2002 and 2019 in two two tertiary Hospitale in Saudi Arabia. The same statment should be reported also in the introducion.
This is an interesting and well written review on very rare lymphomas.
However, I have some concerns as regard the distrubution and diagnosis of the cases. The Authors report 3 cases of CHL (30%) as primary breast lymphomas. This finding is very unusual as extranodal cHL is extremely rare.
On the other hand they report only 1 case of MALT lymphoma (10%) that is much more common in extranodal sites, including the breast.
I encourage the Author to review their diagnosis , considering that some cases of MALT lymphomas may show a high number of CD30 positive cells , even with Reed-Sterberg like morphology, or comment on this.
I would recomand not to use the terminolgy of Burkitt-like for the High grade B Cell , but brather perform FISH for a better characterization.
In the table should also be included some clinica information(i.e. sex, age, responses to therapy)
Author Response
Reviewer 1
Comment 1: The Authors report a series of cases of non-breast implant lymphomas that were diagnosed
as primary breast lymphomas between 2002 and 2019 at two tertiary Hospitals in Saudi
Arabia. The same statement should be reported also in the introduction.
Replay 1: Sure, thank you for
your comment. The statement is reported as advised.
Comment 2: This is an interesting and well written review on very rare lymphomas.
However, I have some concerns as regard the distrubution and diagnosis of the cases. The
Authors report 3 cases of CHL (30%) as primary breast lymphomas. This finding is very
unusual as extranodal cHL is extremely rare.
On the other hand they report only 1 case of MALT lymphoma (10%) that is much more
common in extranodal sites, including the breast.
I encourage the Author to review their diagnosis , considering that some cases of MALT
lymphomas may show a high number of CD30 positive cells , even with Reed-Sterberg like
morphology, or comment on this.
Replay 2: Thank you for your comments, we totally agreed that MALT
lymphoma is common than CHL in this anatomic location. However, all the cases in the
present study were reviewed by expert hematopathologists from two different institutions. The
cells of RS target atypical cells did not show B-cells markers expression except for dim PAX-5
staining and negative CD20, CD97a and other activated B-cells markers.
Comment 3: I would recomand not to use the terminolgy of Burritt-like for the High grade B Cell , but
brather perform FISH for a better characterization.
replay 3: Thank you for your comments, the case
cellular H&E morphology and the tumor size of the cells was larger than the usual Burritt
lymphoma. Also, we avoided to use the term Burritt-like at the time of the diagnosis to avoid
confusion for the clinician. The case was composed of large polygonal cells with high N/C
ration, the size of the cells were larger than the size of Burritt cells. Moreover BCL-2 show
positive tumor expression which was against calling it Burritt lymphoma. But the term was
advised is used in the draft as requested.
Comment 4: In the table should also be included some clinical information(i.e. sex, age, responses to
therapy)
Replay 4: Thank you for your comments, The information of the patient`s age and gender is present
(Table 1) under the table designated as “Summary of the primary breast lymphoma cases
from two tertiary Hospitals in the Western region of Saudi Arabia (n=10)”
Reviewer 2 Report
Comments and Suggestions for Authors
Breast cancer is a leading cause of dead, usually develops from epithelial or stromal cell, the study it is a short report that show a wide range of primary breast lymphoma that can occur in this location, reinforcing the need of conducting additional studies to properly categorize lymphoma subtypes and types due the frequent misidentification, to achieve an accurate diagnosis and ensure providing the appropriate treatment.
As they mention, as this disease displays a range of clinical signs and symptoms, accompanied by different histological and immunohistochemical characteristics; and that each subtype carries a unique prognosis and necessitates specific treatment protocols. To improve the prognostic, differentiate among the histological types for offer an appropriate therapy and increasing the likelihood of successful treatment outcomes:
1.- What strategy do the authors propose to improve the process of diagnosis, detection and classification of primary breast lymphoma?
2.- Since it is a challenge and have a central role in guiding treatment options, what techniques and examinations could validate an early diagnosis?
Author Response
Reviewer 2
Breast cancer is a leading cause of dead, usually develops from epithelial or stromal cell, the
study it is a short report that show a wide range of primary breast lymphoma that can occur in
this location, reinforcing the need of conducting additional studies to properly categorize
lymphoma subtypes and types due the frequent misidentification, to achieve an accurate
diagnosis and ensure providing the appropriate treatment.
As they mention, as this disease displays a range of clinical signs and symptoms,
accompanied by different histological and immunohistochemical characteristics; and that each
subtype carries a unique prognosis and necessitates specific treatment protocols. To improve
the prognostic, differentiate among the histological types for offer an appropriate therapy and
increasing the likelihood of successful treatment outcomes:
Comment 1.- What strategy do the authors propose to improve the process of diagnosis, detection and
classification of primary breast lymphoma?
Replay 1: The study was designated based on two tertiary
centers experience over a long period, spanning over 18 year collections as indicated in the
methodology section. All of these cases were retrieved from the hospitals archives records
and the pathology glassy slides along with their immunohischemistry stains were reviewed by
expert hehatopathologist in the field. Furthermore, the classification and the tumors
designations were obtained according the updated new WHO 20222 Hempath classifications.
Comment 2.- Since it is a challenge and have a central role in guiding treatment options, what
techniques and examinations could validate an early diagnosis?
Replay 2: We totally agree with you, we
believe it is a very rare diagnosis in the breast. However, there are limited number of
publications regarding the role of early detection and since there is no specific sings nor
symptoms for the disease; we totally understand that it is a challenging diagnosis to archives
without the golden slandered of pathology confirmation. We believe that more publications is
needed in this rare entity that is not well established or described in the current literature yet.
Round 2
Reviewer 1 Report
Comments and Suggestions for Authors
I am still concerned about the diagnosis of 3 cases of cHL as primary lymphoma in the breast.
Extranodal CHL is extremely rare and even more in the breast!
Marginal zone lymphoma can have RS-like cell and the differential diagnosis could be very challenging and need a more wide panel of antibodies then those employed by the Authors.